# Implicit Neural Representation Image Codec with Mixed Context for Fast Decoding

## Abstract

Image compression using Implicit Neural Representation (INR) is an emerging technology. While it may not match the quality of cutting-edge autoencoder models, it offers two key benefits: low computational complexity and parameter-free decoding. It also surpasses many traditional and early neural compression methods in terms of quality. In this study, we introduce a new mixed autoregressive model (MARM) to notably decrease the decoding time for the current INR codec, particularly in scenarios with limited computational resources. MARM includes our proposed autoregressive upsampler (ARU) blocks, which are highly computationally efficient, and ARM from previous work to strike a balance between decoding time and reconstruction quality. We also suggest enhancing ARU's performance using a checkerboard two-stage decoding strategy. Moreover, the balance between quality and speed can be adjusted by the ratio of different modules. Comprehensive experiments reveal that our method significantly boosts computational efficiency while preserving image quality. It also significantly excels in decoding acceleration when the quality requirements are more lenient.

## 1 Introduction

Deep learning-based lossy image compression has advanced dramatically in recent years (Ballé et al., 2016; Ballé et al., 2018; Minnen et al., 2018a; He et al., 2022). They have made great progress and outperformed many traditional image codecs such as JPEG (Wallace, 1992) and BPG (Bellard, 2018) on common metrics like PSNR and MS-SSIM (Wang et al., 2003). Joint backward-and-forward adaptive entropy modeling is one of the most important techniques of these models, which makes use of side information in forward adaptation and predictions from causal context of each symbol in backward adaptation (Minnen et al., 2018b; Minnen & Singh, 2020; Cheng et al., 2020). Besides neural image codecs based on autoencoder (AE), the rise of implicit neural representation (INR) in 3D applications, which uses weights of neural network to represent information, has promoted the exploration of using similar technologies in image compression. Dupont et al. (2021a) suggest using 2D coordinates as the input for the MLP and directly outputting the RGB value of the corresponding pixel. Taking it a step further, Ladune et al. (2022) introduce COOL-CHIC, which utilizes trainable latent variables as the input for the MLP.

Although the AE-based methods achieve better rate-distortion performance, INR based methods have three key advantages. The first is low-decoding complexity, which is critical to low-power devices like smartphone or IoT devices. Leguay et al. (2023) achieve a similar BD-rate as HEVC and nearly two magnitude less MACs (Multiplication Accumulation) compared to AE-based methods. The second is parameter-free decoding. Unlike tasks such as classification and generation, the model output in AE-based codec is highly coupled to the model parameters. In other words, the decoding model must have the same structure and set of parameters as the model that generated the compressed representations in an end-to-end compression model. In real-world situations, this character might be problematic. Suppose for some reason we update the parameters of the compression model, such as using a better training dataset. In this case, we must either recompress all of the images that the old model compressed using the new model or save all versions of the model (or diffs of parameters) on the decoder side. Both of these approaches are very resource-intensive. Considering the fact that some users may never update their compression software, updating the compression model is not feasible in practice. Since the network parameters are part of the encoded representation, INR based methods completely circumvent the problem. The last one is INR codec

requires no prior information about image, even the output channels can be specified per image, which is practical for non-RGB pictures like material textures in computer graphics (Vaidyanathan et al., 2023).

However, previous INR-based methods have some shortcomings. Low MACs in COOL-CHIC-like methods do not necessarily imply fast decoding. The main reason is these methods use a pixel-by-pixel fashion to decode the latents, which is suitable for modeling the distribution of pixels but is hard to parallelize Van Den Oord et al. (2016); Van den Oord et al. (2016). Other methods, though do not have these problems, failed to achieve good enough rate-distortion performance Dupont et al. (2021a; 2022); Strümpler et al. (2022). Few of previous works pay attention to both quality and decoding time, which are crucial in practical applications.

In this paper, we focus on improving decoding efficiency while keep reconstruction quality. Our contributions include:

- We propose a parallelization-friendly autoregressive upsampler (ARU) blocks, whcich is highly computationally efficient. The two passes checkerboard strategy in ARU promotes the utilization of context information, improves the reconstruction quality.
- We incorporate ARU blocks and AutoRegressive Model (ARM) in COOL-CHIC to build a a novel Mixed AutoRegressive Model (MARM). The ratio of ARU and ARM is tunable to achieve a more flexible trade-off between quality and speed.
- We propose a new synthesis that combines MLP and CNN to further improve the reconstruction quality.

Thorough experiments over representative datasets were performed, in which our method demonstrates superior efficiency in computational resource-constrained environment while maintaining competitive quality and achieves higher acceleration when relax the quality requirements.

## 2 RELATED WORK

### 2.1 NEURAL IMAGE COMPRESSION

Classical neural image compression methods extend the framework of transform encoding (Goyal, 2001). In this framework, both analysis transform $g_a(\boldsymbol{x}; \phi_g)$ and synthesis transform $g_s(\hat{\boldsymbol{y}}; \theta_g)$ use neural network parametrized by $\phi_g$ and $\theta_g$ as transform functions, rather than linear transforms. In coding procedure, latent representation $\boldsymbol{y}$ generated by $g_a(\boldsymbol{x}; \phi_g)$ is quantized to discrete $\hat{\boldsymbol{y}}$ and losslessly compressed using entropy encoder (Ballé et al., 2016).

The process of quantifying a continues $\boldsymbol{y}$ to a finite set of discrete values will bring problems of information loss and non-differentiable characteristic. The information loss leads to the rate-distortion trade-off

$$\mathcal{L}_{\phi_g, \theta_g} = D + \lambda R. \tag{1}$$

In training stage, the quantization is relaxed by adding standard uniform noise to make the full model differentiable

$$q(\tilde{\boldsymbol{y}}|\boldsymbol{x}, \theta_g) = \prod_i \mathcal{U}(\tilde{y}_i|y_i - \frac{1}{2}, \tilde{y}_i|y_i + \frac{1}{2}). \tag{2}$$

In the framework, the loss function equal to the standard negative evidence lower bound (ELBO) used in variational autoencoder (VAE) training.

There are a lot of papers follow the above framework. Ballé et al. (2018) add scale hyperprior to capture more structure information in latent representation. Minnen et al. (2018a) use an autoregressive and hierarchical context to exploit the probabilistic structure. Minnen & Singh (2020) investigate the inter-channel relation to accelerate the encoding and decoding process. He et al. (2022) use both inner-channel and inter-channel context models and improve the performance.

In these methods, users have to deploy the pre-trained models on both encoding and decoding sides, which may bring problems as depicted in the previous section. But at same time, many insights proposed by these works can also apply to INR-based methods.

## 2.2 Implicit Neural representation

Different from the end-to-end models that use real signals like images or videos as input, implicit neural representation (INR) models generally use coordinates as model input. The network itself is the compressed data representation. This idea thrives on 3D object representation. NeRF (Mildenhall et al., 2020) synthesizes novel views of complex scenes by an underlying continuous volumetric scene function. The function maps the 5D vector-valued input including coordination $(x, y, z)$ and 2D viewing direction $(\theta, \phi)$ to color $(r, g, b)$ and density $\sigma$. MLP is used to approximate the mapping function. To improve model performance, positional encoding is used to enhance visual quality and hierarchical volume sampling is used to accelerate training process.

In addition to NeRF, many insights are proposed by a large body of literature. Park et al. (2019) represent shapes as a learned continuous Signed Distance Function (SDF) from partial and noisy 3D input data. Chen & Zhang (2019) perform binary classification for point in space to identify whether the point is inside the shape. Then the shape could be generated from the result. Müller et al. (2022) proposed to use multi-resolution hash encoding to argument coordinate-based representation and achieve significant acceleration in both training and evaluation without sacrificing the quality.

INR is also used in image-relevant tasks. Chen et al. (2021) extends coordinate-based representation to 2D images and develops a method that can present a picture at arbitrary resolution. Dupont et al. (2021b) propose to generate parameters of the implicit function instead of grid signals such as images in generative models to improve the quality.

Although INR-based methods have succeeded in many areas, popularizing of the technique in compression is non-trivial. The main difference between compression and the tasks above is the model size. In the INR-based compression method, model parameters are also part of the information that needs to be transmitted, which raises the trade-off between model size and reconstruction quality.

## 2.3 INR Based image compression

In image compression, COIN (Dupont et al., 2021a) uses standard coordinate representation that directly maps 2D coordinates $(x, y)$ to color $(r, g, b)$, which allows variable resolution decoding and partial decoding. Along with architecture search and weight quantization to reduce the model size, COIN outperforms JPEG for low bit rate. COIN++ (Dupont et al., 2022) extends the idea of a generative INR method that compresses modulation rather than model weight to achieve data-agnostic compression. In some dataset, COIN++ achieve significant performance improvement.

However, in universal image compression, COIN and COIN++ failed to compete with AE-based neural image codec (Ballé et al., 2018) and JPEG for a high bit rate. Ladune et al. (2022) proposed COOL-CHIC that uses latent along with an autoregressive decoding process to achieve comparable RD performance to AE-based method with low complexity. Leguay et al. (2023) push the performance forward to surpass HEVC in many conditions by leveraging a learnable upsampling module and convolution-based synthesis.

One of the disadvantages of COOL-CHIC-like methods is the theoretical low complexity and slow decoding process because of highly serial decoding process. We propose to replace the ARM model in COOL-CHIC with a parallelization-friendly one to significantly reduce the decoding time.

## 3 Method

### 3.1 System Overview

In image compression task, we define $\boldsymbol{x} \in \mathbb{N}^{C \times H \times W}$ as the $H \times W$ image to be compressed with $C$ channels. For common RGB pictures, $C = 3$. $\hat{\boldsymbol{x}} \in \mathbb{N}^{C \times H \times W}$ is the decoded image. As shown in Fig. 1, our model includes three modules: mixed autoregressive model $f_\psi$, upsampler $f_\phi$ and synthesis $f_\theta$. These networks are parameterized by $\psi$, $\phi$ and $\theta$ respectively. $\hat{\boldsymbol{y}}$ is a set of pyramid-like multi-resolution latent variables with discrete values:

$$\hat{\boldsymbol{y}} = \left\{ \hat{y_i} \in \mathbb{Z}^{H_i \times W_i}, i = 0, 1, \ldots, L-1 \right\}, \text{where} \quad H_i = \frac{H}{2^{L-i-1}}, W_i = \frac{W}{2^{L-i-1}}. \quad (3)$$

Under these notations, the image is $\boldsymbol{x}$ encoded as $\{\psi, \phi, \theta, \hat{\boldsymbol{y}}\}$.

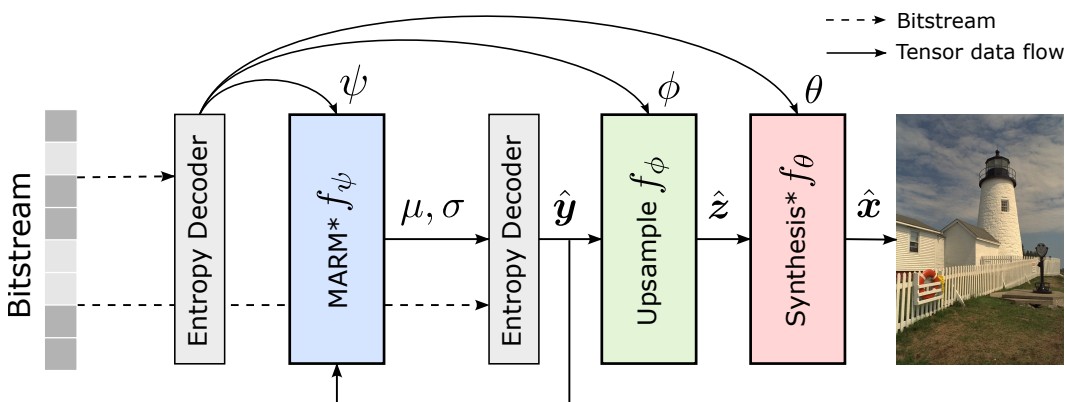

Figure 1: System architecture. The modules mark by * are proposed in this work to replace the original ones. $\psi$, $\phi$, $\theta$ are network parameters for MARM module, upsample module and Synthesis module respectively. These parameters are decoded first to initialize the model. Then the MARM module decodes latents which are integer values matrices with pyramid shapes. Upsample module will transform these latents to a dense representation whtih shape $L \times H \times W$. Syntesis module transforms the dense tensor to image with $(r, g, b)$ channels and $H \times W$ shape. Note since the code length of $\psi, \phi, \theta$ is vary small, we use original notation represents both quantized and unquantized version for simplicity.

When decompressing an image, the first step is decoding the network parameters $\psi$, $\phi$ and $\theta$ and initializing the whole model. Then $\hat{y}$ is decoded from bitstream by $f_\psi$:

$$\hat{y} = f_\psi(b), \tag{4}$$

where $b$ represents bitstream. Like autoencoder codec, $f_\psi$ may have specific structure such as an autoregressive network (Leguay et al., 2023). Because making use of predictions from causal context of each symbol in this stage is very important to remove redundancy and reduce bit rate (Ballé et al., 2018; Minnen et al., 2018a; Minnen & Singh, 2020; Cheng et al., 2020). After that, a dense representation $\hat{z} \in \mathbb{R}^{L \times H \times W}$ is obtained by the learnable upsampler $f_\phi$:

$$\hat{z} = f_\phi(\hat{y}). \tag{5}$$

Finally, decoded image $\hat{x}$ is reconstructed from $\hat{z}$

$$\hat{x} = f_\theta(\hat{z}). \tag{6}$$

Previous work have investigated the performance of full MLP and full convolutional network as synthesis. However the prior enforced by both structure may not apply for all input data. To enhance the generality of method, we combine MLP and convolution layer by a residual connection, as shown in Appendix A.

For encoding stage, different from AE-based neural image codec, implicit neural representation based neural image codec does not require encoder. The encoding process of such methods is the process of training neural networks. Although the coding process is different, the final target function is the same:

$$\mathcal{L} = D(x, \hat{x}) + \lambda R(\hat{y}), \tag{7}$$

where $D$ is distortion function such as mean squared error and $R$ approximate rate with entropy. Since the discrete value $\hat{y}$ is non-differentiable, which is common to deep-learning compressor, we use a set of real value $y$ with same shape as $\hat{y}$ and a quantization function $Q$ in training

$$\hat{y} = Q(y). \tag{8}$$

$Q$ could be either a fixed uniform scalar quantizer (Ballé et al., 2016) or $\epsilon$-STE quantizer (Leguay et al., 2023) according to training stage.

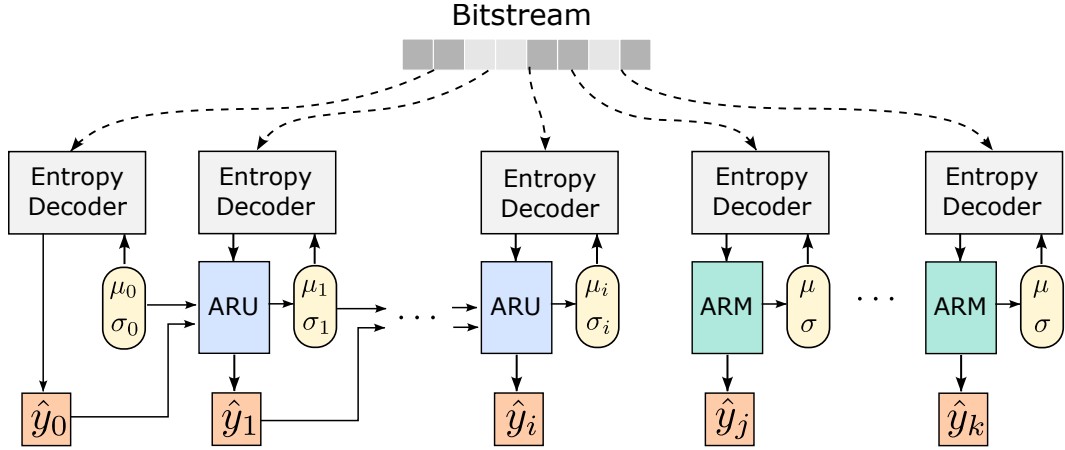

Figure 2: Network architecture of the MARM. Left part is channel-wise autoregressive model, which predict entropy parameters conditioned on previous latent. Right part correspond to inner-channel auto-regression model generation parameters pixel by pixel. Given the number of ARU blocks $M$ and total latents number $L$, we note $i = M - 1$, $j = M$ and $k = L - 1$ for simplicity. $\mu_0$ and $\sigma_0$ are initialized to tensor with 0 and $\exp(-0.5)$ respectively for all images. $\mu$ and $\sigma$ with subscription means the values output as a matrix rather than serially generated scalar for those without subscription.

## 3.2 Mixed Autoregressive Model

Autoregressive network is widely used in casual context prediction (Minnen et al., 2018a; Leguay et al., 2023), which demonstrates the effectiveness of the structure in reducing redundancy of compressed representation. This is more clear if we decompose the second term of Eq. 7

$$R(\hat{\boldsymbol{y}}) = D_{KL}(q||p_\psi) + H(\hat{\boldsymbol{y}}), \tag{9}$$

where $D_{KL}$ stands for the Kullback-Leibler divergence and $H$ for Shannon's entropy. The first term suggest the closer we approximate to real distribution $p$, the more bit we will save. In ARM model, $p_\psi$ is decomposed as:

$$p_\psi(\hat{\boldsymbol{y}}) = \prod_{l,i} p_\psi(\hat{y}_{l,i}|\mu_{l,i}, \sigma_{l,i}), \text{where } \mu_{l,i}, \sigma_{l,i} = f_\psi(\hat{y}_{l,<i}), \tag{10}$$

where $l$ means the $l$-th latent and $< i$ means all pixels in a flatten latent whose index is smaller than $i$. Obviously, the decoding proceeds pixel by pixel, which is time consuming and hard to parallelize. To alleviate such problem, our autoregressive upsampler (ARU) apply autoregressive decoding across latents. In other word, we use low-resolution latent to predict the decoding parameter of next high-resolution latent:

$$p_\psi(\hat{\boldsymbol{y}}) = \prod_{l} p_\psi(\hat{y}_l|\mu_l, \sigma_l), \text{where } \mu_l, \sigma_l = f_\psi(\hat{y}_{l-1}, \mu_{l-1}, \sigma_{l-1}). \tag{11}$$

This approach can significantly improve the parallelism of the autoregressive module and greatly enhance the computational performance. Similar technique is also used in some previous work (Reed et al., 2017).

While ARU block outperforms in efficiency, ARM can recognized more correlation between adjacent pixels because of locality inside each latent. So we integrate ARU and ARM to a Mixed AutoRegressive Model (MARM), as shown in Fig. 2. For low-resolution latents, which have more global information, ARU is used to accelerate decoding process. For high-resolution latents, we use ARM to capture more details such as textures. The ratio of two type blocks is controlled by a hyperparameter $M$, which means the number of ARU blocks in MARM. Note when $M = 0$, the MARM becomes ARM.

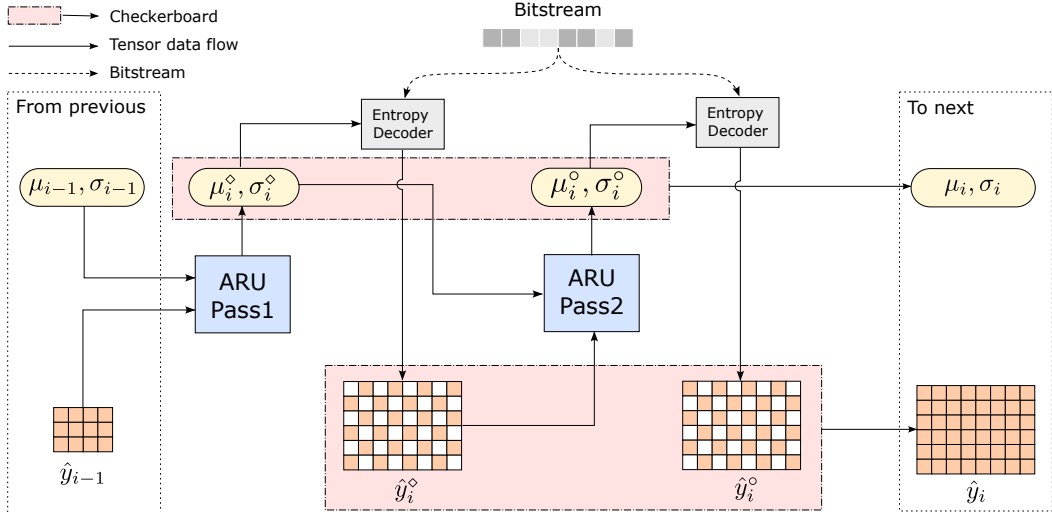

Figure 3: Network architecture of ARU. ARU uses previous parameter matrix $\mu_{i-1}, \sigma_{i-1}$ and latents as input. ARU pass1 output decoding parameters $\mu_i^\diamond, \sigma_i^\diamond$ for anchors in latents. ARU Pass2 generate parameters for non-anchors conditioned on $\mu_i^\diamond, \sigma_i^\diamond$ and anchors. Checkerboard means to merge anchor of the first tensor and non-anchor of the second tensor. Detailed structure of ARU Pass1 and ARU Pass2 can be found in appendix

### 3.3 TWO STAGES ARU

Although using low-resolution latent to predict higher-resolution ones is target to improve computational performance at the cost of reconstruction quality, the degradation can be reduced. Different from pixel-by-pixel correlation or cross resolution correlation, we can utilize the locality in only two pass in a checkerboard fashion. As shown in Fig. 3, we mark anchor in tensor $\hat{y}_i$, $\mu_i$, $\sigma_i$ (orange ones of $\hat{y}_i^\diamond$ in Fig.3 ) as $\hat{y}_i^\diamond, \mu_i^\diamond, \sigma_i^\diamond$, non-anchor (white ones of $\hat{y}_i^\diamond$ in Fig.3 ) as $\hat{y}_i^\circ, \mu_i^\circ, \sigma_i^\circ$ respectively. Following the notation, joint distribution of $\hat{y}_i$ can be written as

$$p_\psi(\hat{y}_i|\mu_i, \sigma_i) = p_\psi(\hat{y}_i^\diamond|\mu_i, \sigma_i) \cdot p_\psi(\hat{y}_i^\circ|\hat{y}_i^\diamond, \mu_i, \sigma_i) = p_\psi(\hat{y}_i^\diamond|\mu_i^\diamond, \sigma_i^\diamond) \cdot p_\psi(\hat{y}_i^\circ|\mu_i^\circ, \sigma_i^\circ). \quad (12)$$

The anchor pixels only depend on information from previous low-resolution latent, and the correlation is fitted by $f_\psi^\diamond$:

$$\mu_i^\diamond, \sigma_i^\diamond = f_\psi^\diamond(\hat{y}_{i-1}, \mu_{i-1}, \sigma_{i-1}). \quad (13)$$

For decoding of non-anchor pixels, all previous information is available, including the decoded value of anchor $\hat{y}_i^\diamond$. As another form of making use of causal context information, ARU Pass2 can compute $\mu_i^\circ$ and $\sigma_i^\circ$ accordingly

$$\mu_i^\circ, \sigma_i^\circ = f_\psi^\circ(\hat{y}_i^\diamond, \mu_i^\diamond, \sigma_i^\diamond). \quad (14)$$

The idea of parallel predicting probability mass function of compressed representations have been investigated in some AE-based image or video codec (He et al., 2021; Li et al., 2023). Same as these method, INR codec also benefits from this design.

### 3.4 COMPLEXITY ANALYSIS

In INR codec, the process of decoding latent takes part majority of decoding time in many cases. Given a image with $n = H \times W$ pixels, the total number latent pixels need to be decoded is $O(n)$. Because of serial decoding, time complexity is the same.

If we suppose parallel operations such as convolution operation over a feature map can be finished at $O(1)$, which is practical for not very large pictures in even low-power device with SIMD support, the decoding time complexity of MARM is $O(M \log n + (L-M)n)$. When $M = L$, the complexity of our method becomes $O(\log n)$, which surpass the previous work. When $M < L$, the complexity is similar to previous work. But in realistic setting, $n$ is finite, which means the constant factor is important as well. Actually, experiments support when $L - M \leq 2$, the acceleration is still significant.

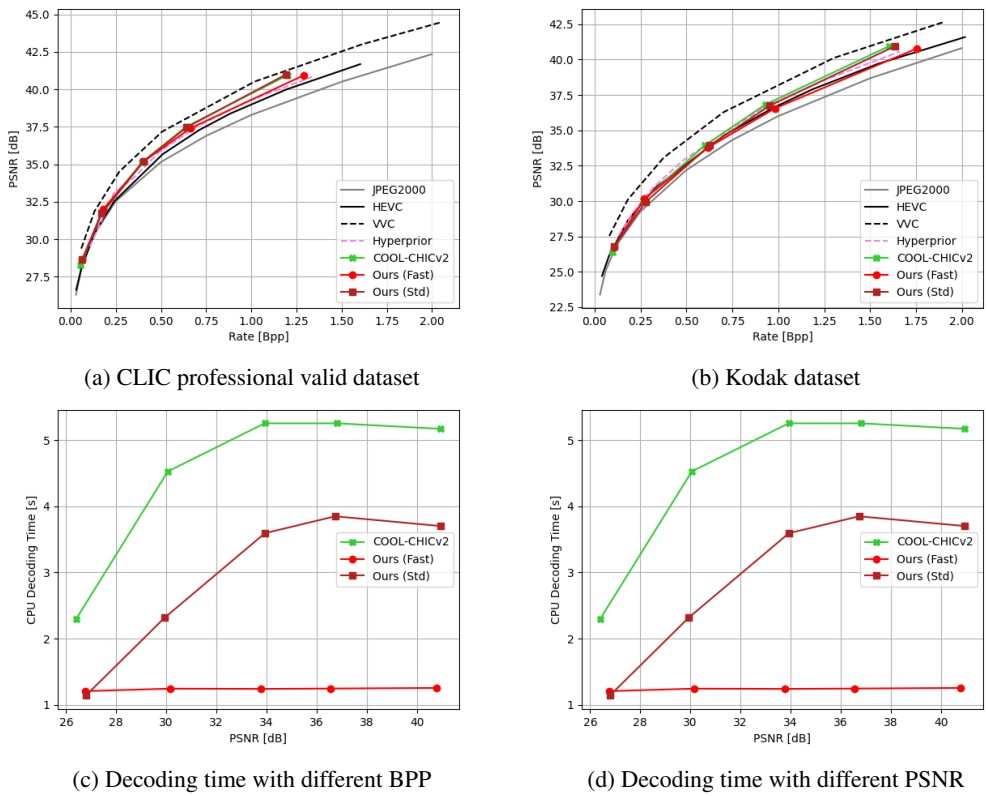

(a) CLIC professional valid dataset

(b) Kodak dataset

(c) Decoding time with different BPP

(d) Decoding time with different PSNR

Figure 4: Main results of experiments. Fig. 4b and Fig. 4a shows the rate-distortion performance averaged over CLIC professional valid dataset and Kodak dataset respectively. Fig. 4c and Fig. 4d shows the decoding time averaged over kodak dataset as function of BPP and PSNR. All COOL-CHIC-like methods aggregate the result over same $\lambda$. We observe our standard (mark as std) method achieve comparable reconstruction quality with COOL-CHICv2 on both kodak dataset and CLIC dataset. At the same time, we reduce time consumption of decoding on CPU by a large margin. If we relax quality restriction to set $M = L$ i.e. using pure ARU blocks in MARM module, the acceleration increases significantly.

## 4 EXPERIMENT

### 4.1 DATASETS AND EXPERIMENTS SETUP

The experiment use image from CLIC professional valid set [1] and Kodak dataset (Kodak, 1993). CLIC dataset contains a collections of natural images with different resolution. Since training set in unnecessary for INR codec, we only perform experiments on valid set. Kodak dataset is another widely used dataset in image compression community, which includes 24 images of size $768 \times 512$. The main experiment results and more ablations are performed in this dataset.

To ensure fairness in comparison, all learning-based model is implemented using PyTorch without special optimization. For AE-based models, we use the pre-trained model in CompressAI (Bégaint et al., 2020). Our model is implemented based on previous work of Leguay et al. (2023), which use constriction package (Bamler, 2022) as entropy encoder. We note the work as COOL-CHICv2 in the rest of the paper. Following setup of COOL-CHICv2, we use the configurations of $L = 7$, $M = 5$ as our standard model and $L = 7$, $M = 0$ for fast model. The model is trained on each of images in dataset, using the loss function presented in eq.7.

---

[1]https://clic.compression.cc/2021/tasks/index.html

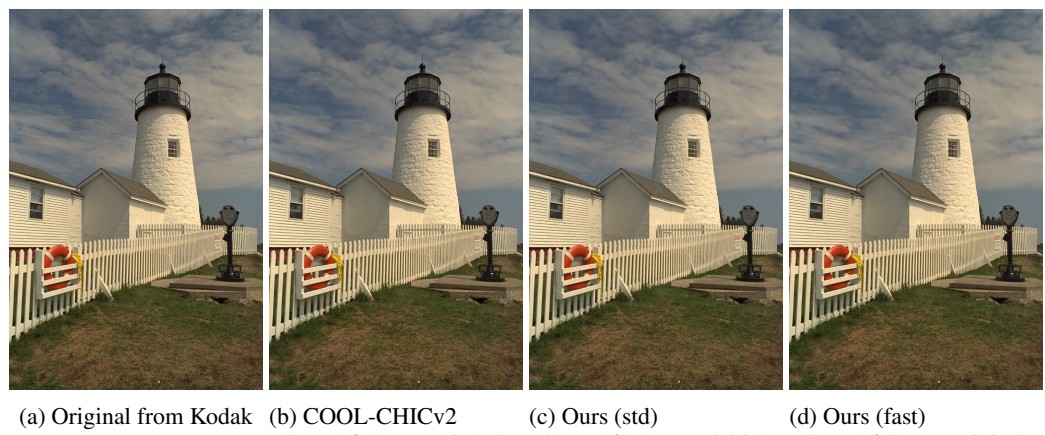

(a) Original from Kodak  (b) COOL-CHICv2  (c) Ours (std)  (d) Ours (fast)
PSNR 36.82, BPP 0.874  PSNR 36.82, BPP 0.905  PSNR 36.42, BPP 0.974

Figure 5: Visualization of decoded images

## 4.2 QUALITY RESULTS AND COMPUTATIONAL PERFORMANCE

We use peak signal-to-noise ratios (PSNR) as quality measurement and bit-per pixel (BPP) as coding efficiency metric. Fig. 4a and Fig. 4b illustrate the decompression results of our method. Fig. 5 shows the quality results. Not surprisingly, our method performs well on metric of reconstruction quality while fast method exceeds all COOL-CHIC-like method on decoding time, according to Fig. 4c and 4d. COOL-CHICv2 and our method achieve faster decoding time at low bit rate because of all zero high-resolution latents, which means for these latents codec only need to transmit a special tag rather than decode them.

## 4.3 COMPOSITION OF MARM

As mentioned before, $M$ controls the ratio of ARU blocks and ARM blocks, and model performance is highly correlated to the parameter. Fig. 6a illustrates the overall decoding quality and efficiency. To comprehensively evaluate the efficiency of a codec, we suggest to use the Time BD-rate (TBD-rate), which is a variant of BD-rate as the measurement. When calculating TBD-rate, We only need to replace BPP with decoding time in common BD-rate formula.

The trend of TBD-rate is easy to understand. More ARU block lead to more significant acceleration. But the quality does not improve when we increase the ratio of ARM blocks. In our experiment, the model reaches the best quality at $L - M = 3$. When number of ARM keep increasing, the performance degrade drastically. This may caused by the correlation between latents. For high-resolution latent, ARU depend less on previous latent while for low-resolution latents inter-latent is important. On the contrary, ARM does not depend on inter-latent information. Since we generally use large $M$ in our method, this phenomenon will not affect the effectiveness of our method.

## 4.4 MORE ABLATIONS

Fig.7b illustrate the performance of checkerboard and our proposed new synthesis module. No checkerboard means is we omit the second pass when decoding latent i.e. use $\mu_i^\diamond, \sigma_i^\diamond$ directly to decode both anchors and no-anchors, Old synthesis means we use the original one in COOL-CHICv2 It is obvious that these two structure further improve the RD-performance. Fig. 7a shows the bit allocation. We observe that ARM blocks will allocate more bit to high-resolution latents compare to ARU blocks, which indicates the inter-latent correlation.

## 5 DISCUSSION

This work presented a new module MARM to enhance the current implicit neural representation image codec, which is the first method that achieves comparable performance in both reconstruction

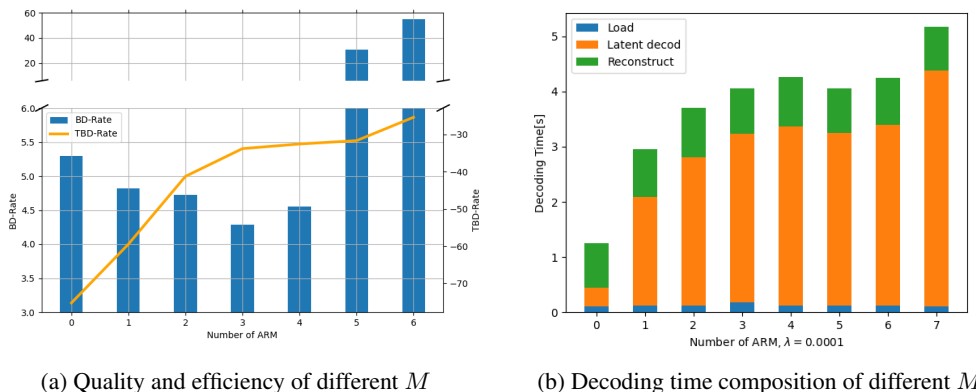

(a) Quality and efficiency of different $M$

(b) Decoding time composition of different $M$

Figure 6: Ablation results when $M$ differs. Fig. 6a shows the BD-rate and TBD-rate for different $M$ average over Kodak dataset. Fig. 6b shows the composition of decoding time. For more result see Appendix. B

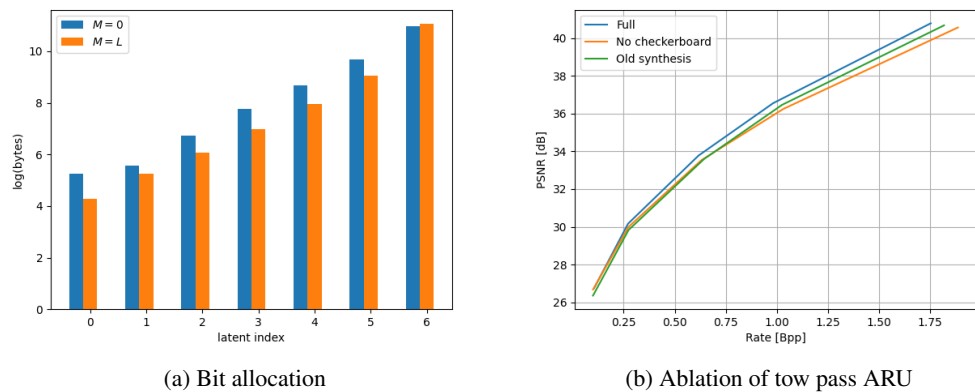

(a) Bit allocation

(b) Ablation of tow pass ARU

Figure 7: Ablation results. Fig. 7a shows the bits allocation among latents when $M = 0$ and $M = L$, correspond to MARM with full ARM blocks and full ARU blocks respectively. 7b shows the rate-distortion curve average over kodak dataset with different settings when $M = L = 7$.

quality and decoding time. The MARM module improves the computation efficiency by leveraging the channel-wise autoregressive architecture in low-resolution latent and pixel-wise autoregressive to maintain the decompressing quality. The experiments show that our modificationreduce the decoding time by a large margin without significant quality degradation.

Although we improve the current INR-based image codec, there are still some limitations. The first is decompressed images failed to achieve the comparable quality in both PSNR and MS-SSIM simultaneously when using MSE distortion metric. Model capacity which is constrained by model size can be a possible cause. In the COOL-CHIC-like method, the majority of bits are allocated to latents while only a small part is to the network. A very small network makes it hard to capture complex patterns. How the bits allocating ratio of latents and network affects model performance is still an open problem. The second is our method can not be significantly accelerated by GPU. The main reason is the coupling between entropy decoding and parameter estimation, which is more like serial operation rather than parallel computation. There are still many engineering problem to solve. The third problem is the high time complexity of encoding process. This problem exists in all INR-based methods. Although there are some acceleration methods, extending these methods to image compression is not easy. Possible reason is that most 3D representation is highly sparse comparing to 3D space grid, while most of images such as natural photos are dense signals in spatial domain. Developing new method or adapting above methods to accelerate encoding process is crucial for INR-based image codec. These limitations are also great opportunities. We will therefore further investigate these in future work.

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

# A  NETWORK STRUCTURE

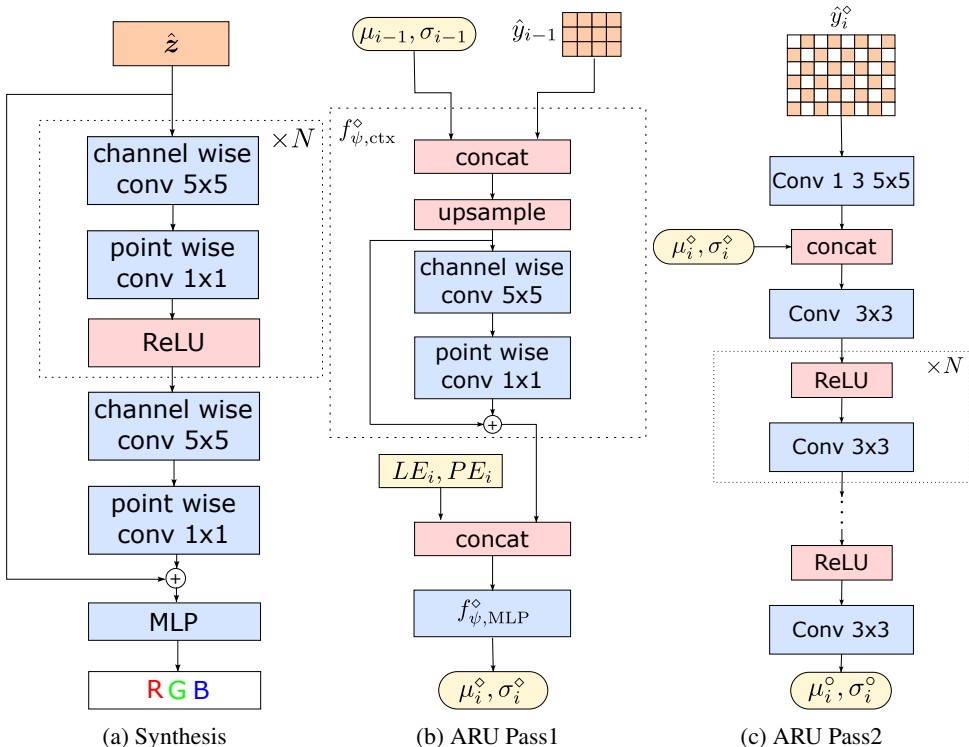

(a) Synthesis  (b) ARU Pass1  (c) ARU Pass2

Figure 8: Network structure

## A.1  NEW SYNTHESIS

Previous works have investigated full MLP synthesis and full CNN synthesis. The techniques like residual connection are also applied in block level. In this work, we propose a mixed synthesis with a residual connection from dense representation to final MLP. The network can balance the information from both residual connection and from CNN blocks,as shown in Fig. 8a.

## A.2  TWO PASS ARU

The detailed structure of two ARU sub-blocks is show in Fig. 8b and 8c. For simplicity, we only introduce the overall data flow of the module. In addition to previous content, we add level encoding and positional encoding to promote the networks capability. As shown in 8b, Eq. 13 can be decomposed to three steps:

$$v_{\text{ctx}} = f^{\diamond}_{\psi,\text{ctx}}(\hat{y}_{i-1}, \mu_{i-1}, \sigma_{i-1}), \tag{15}$$

$$v^{\diamond}_{i,ab} = \text{Concat}([v_{\text{ctx},ab}, PE_{i,ab}, LE_{i,ab}]), \tag{16}$$

$$\mu^{\diamond}_{i,ab}, \sigma^{\diamond}_{i,ab} = f^{\diamond}_{\psi,\text{MLP}}(v^{\diamond}_{i,ab}). \tag{17}$$

To reduce the dimention of the output, we only use a simple mesh grid positional encoding. Let $\{(a,b) \in \mathbb{Z}^2, 0 \le a < H_i, 0 \le b < W_i - 1\}$ represents spatial location in $i$-th latent, the position encoding is defined as

$$PE_{i,ab} = [\frac{a}{H_i} - 0.5, \frac{b}{W_i} - 0.5]. \tag{18}$$

And the level encoding is defined as

$$LE_{i,ab} = \frac{2i}{L}. \tag{19}$$

# B    DECODING TIME COMPOSITION

Fig. 9 and Fig. 6b illustrate the detailed decoding time composition with different $M$. Obviously, for allow parameters setting, the decoding time of latents dominates the decoding efficiency, which is the main focus of our work. Note the loading time cannot be omitted or amortized since the model weights are part of compressed representation for each image.

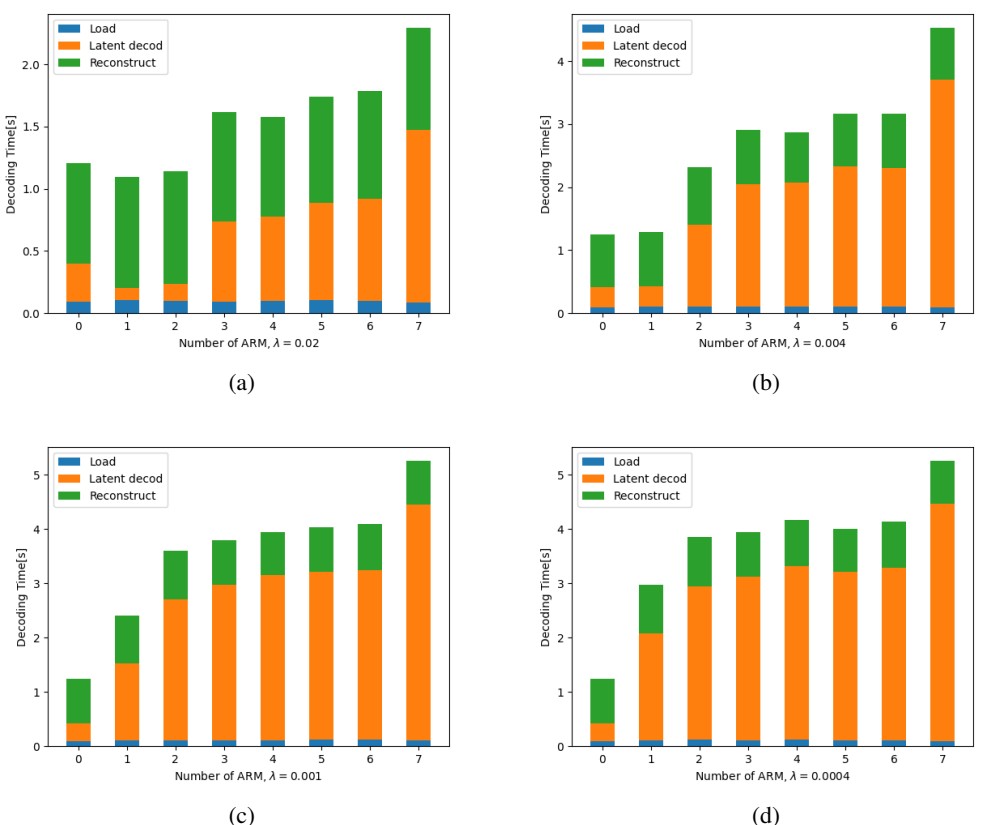

Figure 9: Decoding time composition when $\lambda$ is different.

# C    BIT ALLOCATION

One of the most attractive characters of COOL-CHIC-like methods is the underlying bit allocation mechinism. Fig. 10 - 14 are the allocation results on Kodak dataset for different $M$ and $\lambda$. Not superisingly, majority of bits is allocate to the latents when $\lambda$ is small, which means more information of details is preserved. When $\lambda$ becomes larger, the codec chooses to allocate more bits to coarser latent, as shown in Fig. 13 and Fig. 14.

An exception occurs when $M = 2$, which is also shown in Fig. 6a. As discussed before, this may cause by confliction of utilizing inter-latent information, so we need to set proper $M$ in practical applications.

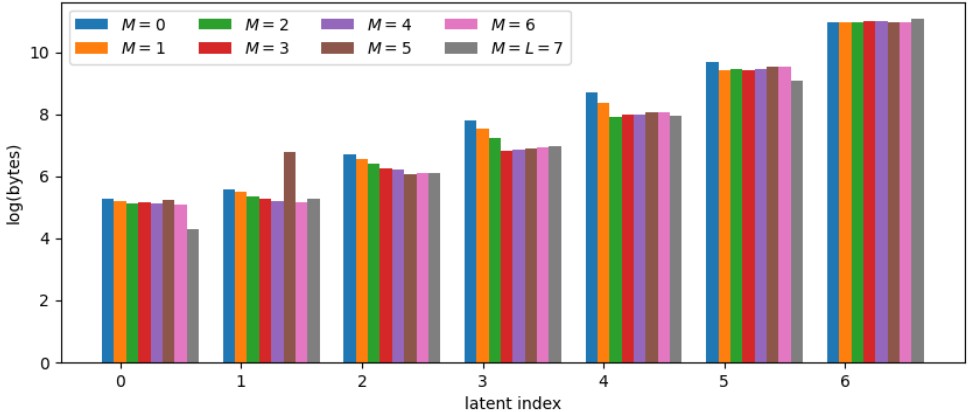

Figure 10: $\lambda = 0.0001$

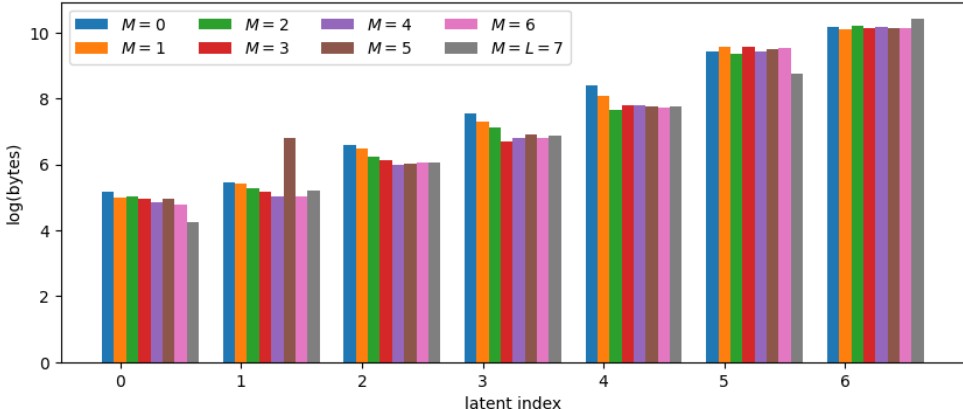

Figure 11: $\lambda = 0.0004$

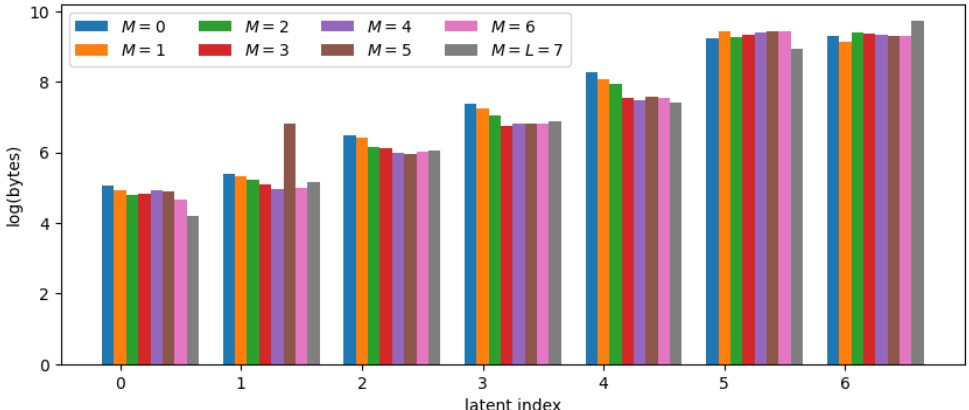

Figure 12: $\lambda = 0.001$

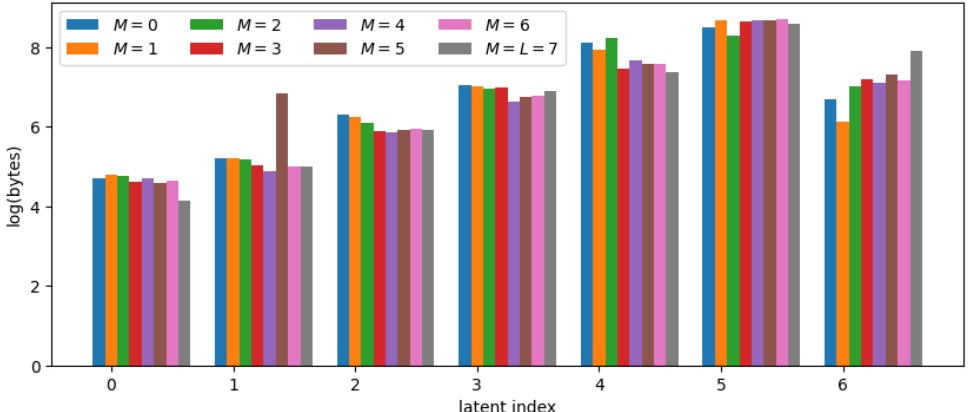

Figure 13: $\lambda = 0.004$

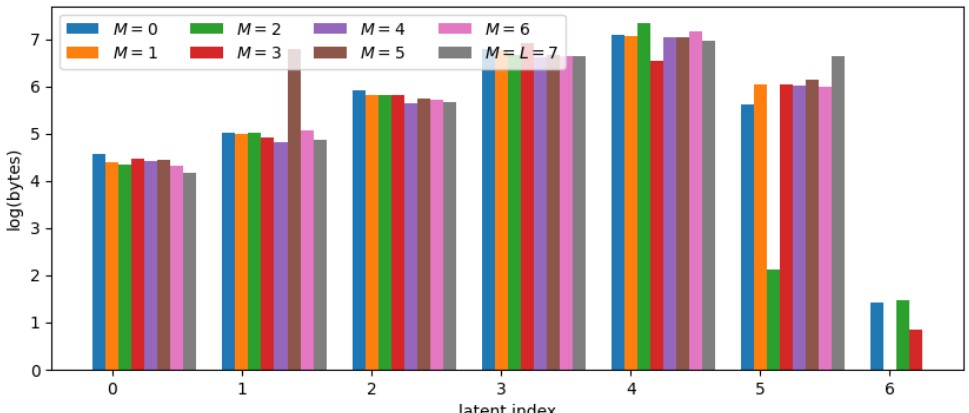

Figure 14: $\lambda = 0.002$

