# OpenReview forum: "Implicit Neural Representation Image Codec with Mixed Context for Fast Decoding"
_ICLR.cc/2024/Conference — ICLR 2024 Conference Withdrawn Submission_

### Official Review · Reviewer_8Lyx · 2023-10-29

**Soundness:** 2 fair
**Presentation:** 2 fair
**Contribution:** 2 fair
**Rating:** 3
**Confidence:** 4

**Summary:**

The paper proposes to speed up the decoding process of implicit neural representations by introducing a two pass autoregressive formulation that allows parallelization during implementation using a checkerboard processing scheme. The method demonstrates improvements over the COOL-CHIC method, which is a low complexity implicit neural coding approach.

**Strengths:**

- It tackles an interesting topic, i.e. efficiency of INR codecs.
- It proposes an implementation trick to improve the decoding time of implicit neural representations for images.
- It demonstrates interesting results in terms of decode time!

**Weaknesses:**

- the paper is an incremental contribution on COOL-CHIC method.
- the paper is limited to the empirical implementation aspects which could improve a previously presented method.
- the paper is slightly difficult to follow at some sections. For example, the connection between ARU and ARM is a bit unclear and figure 2 seems to add a bit to the confusion around the connection between ARU and ARM.
- the paper may benefit from proof reading and checking the definitions of the parameters in formulas, e.g., section 3.2 may benefit from definitions of parameters. (albeit some are previously defined, and some could be guessed)
- the paper lacks proper evaluation against other INR based methods, e.g., the video-based approaches that are discussed to be efficient in all-intra mode, a good example could be "HNeRV: A Hybrid Neural Representation for Videos", (CVPR 2023).
- the results in figure 4, given the  complexity analysis and the configuration of M and L, seems contradicting the statement in section 3.4, "When M < L, the complexity is similar to previous work" whereas the value of M = 0 and L = 7 and the method outperforms the previous methods in terms of decode time with a significant margin!

**Questions:**

- putting the M=0 results in the fast model, where the decode time is almost constant for various rates and PSNR and significantly faster than baseline, why?  according to complexity analysis, shouldn't M and L be equal to get the best results?
- why there is no report for M = L within the configurations? Isn't it supposed to be the best performing configuration, in terms of decode time?

---

### Official Review · Reviewer_GVAi · 2023-10-31

**Soundness:** 3 good
**Presentation:** 3 good
**Contribution:** 3 good
**Rating:** 6
**Confidence:** 4

**Summary:**

This paper presents a mixed autoregressive model (MARM) designed to significantly reduce the decoding time of the current INR codec. The MARM incorporates autoregressive up-sampler (ARU) blocks. These blocks are designed to be computationally efficient, adjusting trade-off between fast decoding and high-quality reconstruction. The method includes a checkerboard two-stage decoding approach to enhance the ARU’s process.

**Strengths:**

The proposed solution is computationally efficient. This is particularly important for real-world applications, when considering the INR-based codec has gained considerable attention in the fields of image compression. The methods seem to offer a good trade-off between decoding speed and image quality, by optimizing the structures and enhancing parallelization. Those properties are important in modern decoders to process high-quality and high-resolution images.

**Weaknesses:**

There are several questions on experimental results.

1. The BD-rate is significantly increasing, when the number of ARM is greater than or equal to 5. How a practical decoder determines the number in practice?

2.  What are CPU decoding time in Fig. 4 and decoding time in Fig. 6. General image codec (e.g. JPEG 2000) is much faster. Although the proposed method significantly reduces the decoding time, it seems to be much far from a practical decoder, when observing the numbers.

3. Although the design of the decoder seems to generally offer a good trade-off, the worst-case scenarios should be analyzed in the ablation tests, in terms of quality degradation, speed, etc, and their reasons. Furthermore, the proposed method would degrade a throughput of a decoder. However, the paper does not address the issues in experiments.

**Questions:**

The reviewer looks forward to hearing the authors' response about the questions above.

---

### Official Review · Reviewer_wVzy · 2023-11-01

**Soundness:** 2 fair
**Presentation:** 3 good
**Contribution:** 2 fair
**Rating:** 3
**Confidence:** 4

**Summary:**

The authors present a mixed autoregressive model (MARM) for accelerating the entropy decoding process of the current INR codec. To relax the computational burden of the pixel-wise spatial autoregressive model, the MARM introduces channel-wise autoregressive structure in low-resolution latent space based on the proposed autoregressive upsampler (ARU) blocks. They propose a two-pass checkerboard strategy in ARU and a new synthesis layer to further improve the reconstruction quality.

**Strengths:**

Significantly reducing the decoding complexity with minor performance drop.
The writing is clear and easy to follow.

**Weaknesses:**

1.  One concern is about motivation, the importance of reducing decoding time of INR-based codec. The decoding complexity of INR-based codec is already orders of magnitude less than AE-based codec. The bottleneck of INR-based codecs in practical applications mainly lies in their excessively high encoding complexity. Therefore, I think the decoding time problem is relatively less significant.
2. Moreover, novelty of the proposed method is somewhat incremental. The concept of channel-wise autoregressive model [1] and checkboard context model [2] and their combination (ELIC) [3] have already appeared in previous neural image codecs. In this paper, the structure of the proposed two-stage ARU is very similar to the structure of spatial-channel contextual adaptive model in ELIC [3]. Although it may cost a significant amount of effort (e.g. engineering effort) to introduce such kind of entropy model in INR-based codecs, the novelty of this paper is probably limited to some extent.

[1] Channel-wise autoregressive entropy models for learned image compression. ICIP 2020
[2] Checkerboard context model for efficient learned image compression. CVPR 2021
[3] ELIC: Efficient Learned Image Compression with Unevenly Grouped Space-Channel Contextual Adaptive Coding. CVPR 2022

**Questions:**

(1)	In Fig. 6(a), it shows that the R-D performance drops dramatically when the number of ARM increases from 4 to 6. However, the entropy models used in the anchor model COOL-CHIC are all ARMs. Why COOL-CHIC still achieves the SOTA performance?
(2)	As mentioned in the paper, INR-based image codecs are much faster than AE-based image codecs in decoding process, but are slower in encoding process. I was wondering how fast it will be for a practical INR-based codec. Could you provide the decoding/encoding time comparison on GPU/CPU?